# STEALTHY TARGETED BACKDOOR ATTACKS AGAINST IMAGE CAPTIONING

## ABSTRACT

We study backdoor attacks against image caption models, whose security issues have received less scrutiny compared with other multimodal tasks. Existing backdoor attacks typically pair a trigger either with a predefined sentence or a single word as the targeted output, yet they are unrelated to the image content, making them easily noticeable as anomalies by humans. In this paper, we present a novel method to craft targeted backdoor attacks against image caption models, which are designed to be stealthier than prior attacks. Specifically, our method first learns a special trigger by leveraging universal perturbation techniques for object detection, then places the learned trigger in the center of some specific source object and modifies the corresponding object name in the output caption to a predefined target name. During the prediction phase, the caption produced by the backdoored model for input images with the trigger can accurately convey the semantic information of the rest of the whole image, while incorrectly recognizing the source object as the predefined target. Extensive experiments demonstrate that our approach can achieve a high attack success rate while having a negligible impact on model clean performance. In addition, we show our method is stealthy in that the produced backdoor samples are indistinguishable from clean samples in both image and text domains, which can successfully bypass existing backdoor defenses, highlighting the need for better defensive mechanisms against such stealthy backdoor attacks.

## 1 INTRODUCTION

Deep neural networks (DNNs) have achieved state-of-the-art performance on many machine learning tasks such as computer vision (CV) (He et al., 2016) and natural language processing (NLP) (Bahdanau et al., 2014; Pratap et al., 2019). Despite achieving tremendous success, DNNs are highly vulnerable to adversarial attacks, including both testing-time attacks such as adversarial examples (Szegedy et al., 2014; Goodfellow et al., 2015) and training-time attacks such as backdoor attacks (Gu et al., 2019). Among them, backdoor attacks are particularly harmful due to their stealthiness, as a backdoored model functions normally on clean data but only behaves abnormally when exposed to data crafted with a specific trigger. The ease of implementation and difficulty of detection enable them to be one of the mainstream studied attacks in deep learning. Backdoor attacks are primarily studied for CV and NLP tasks, however, with the emergence of multimodal models, recent studies have shifted their focus towards exploring backdoor attacks for multimodal tasks, such as text-to-image generation (Struppek et al., 2022) and visual question answering (Walmer et al., 2022). Compared with single-domain scenarios, successfully launching backdoor attacks on multimodal models can be more challenging, because the attacker needs to consider the interactions and relationships among multiple data types.

Nevertheless, image caption models, a specific type of multimodel models, have garnered scant attention in terms of their security vulnerabilities. Previous backdoor attacks on image caption models (Li et al., 2022; Kwon & Lee) have associated the trigger with a specific sentence or single word. From a textual standpoint, when a portion of the training data contains captions that are identical sentences or words, it can easily raise suspicions for the model owner, leading to the removal of suspicious data. From a visual perspective of stealthiness, the attacker needs to carefully craft the poisoned samples with triggers indistinguishable from clean data. Therefore, it remains elusive whether these existing backdoor attacks are useful if the victim employs the aforementioned defense

mechanisms before training the image caption model. To achieve concealment for backdoor attacks on image caption models across both data modalities, we aim to explore the possibility of establishing a connection between the trigger and a specific object within the sentence, rather than the entire sentence.

**Contributions.** In this paper, we propose a novel targeted attack to generate stealthy backdoored samples for image caption models. The adversary aims to poison the victim model to produce captions with incorrectly-specified target name of certain source object, whenever the model encounters an input image with some special trigger placed on the source object (Section 3). In particular, we utilize an object detection model to generate a universal adversarial perturbation as the trigger, since we discover that randomly generating a trigger pattern and forcibly associating it with the target object will decrease the model's standard functionality (Section 3.1). Our solution is to optimize the trigger pattern in a way that the features of the source object, when overlaid with the trigger, become similar to the features of the target object. Unlike in the image domain, where the backdoor trigger is usually placed to the same location across images, we add the trigger to the center of the bounding box of the selected source object. As a result, the trigger location will differ from image to image, thereby increasing the hardness for the backdoored samples to be detected. In the meantime, we change the name of the source object that appears in the caption to the target object name chosen by the adversary, which also increases the difficulty of detection, since the modified image captions are supposed to have minimal changes compared with the ground-truth and may vary for different backdoor samples. To ensure the concealment of the produced trigger, we enforce the small $l_\infty$-norm constraint during its generation (Section 4.1). Moreover, to enable the model to learn the association between the trigger and the target object name more effectively, we propose to inject both backdoor samples and their corresponding clean ones with the trigger removed (Section 4.2). Extensive experiments on benchmark datasets demonstrate that our proposed method can achieve over 90% attack success rate while having a minor impact on clean performance (Section 5.1). To further examine the stealthiness of our attack, we test the existing backdoor defense methods, and the results show that our method can successfully bypass these defenses (Section 5.2).

## 2 RELATED WORK

**Image Captioning.** The goal is to learn an image caption model to produce sentences to properly describe the content for any given input image. To achieve such goal, the image caption model needs to be trained to first understand the feature information of the image and the relationship between the various objects in the image then convert the image features into corresponding human language. In particular, Vinyals et al. (2015) proposed to leverage both convolutional neural networks (CNNs) and long short-term memory networks (LSTMs), where image features are first extracted by a pre-trained GoogLeNet then averaged and fed into an LSTM to predict the image caption. Subsequently, Xu et al. (2015) introduced the attention module to automatically build image caption models, enabling the model to pay more attention to image regions related to the words since the image caption is generated word by word. More recently, Huang et al. (2019) improved the functionality of attention-based models by proposing the *Attention on Attention* module to measure the correlation between value and query, while Wang et al. (2023) introduced the *Multiway Transformer* for generic modeling, which yielded deep fusion and modality-specific encoding. In this work, we consider the model proposed by Xu et al. (2015) as used by the victim, since most of state-of-the-art image caption models adopt the similar encoder-decoder framework and attention mechanisms.

**Backdoor Attacks.** Backdoor attacks are mostly studied in the context of CV and NLP tasks. Gu et al. (2019) proposed the first backdoor attack against DNN models for image classification tasks, where the adversary crafted some special triggers like black and white squares to the training data and changes the underlying class label to some target label to mount the poisoning attack. When the model is trained with such poisoned samples, a test image crafted with a similar trigger will be predicted as the target label. More recent research has made efforts in enhancing the stealthiness of backdoor attacks in image domains. These improvements involve constraining pixel differences (Chen et al., 2017; Liao et al., 2018), ensuring consistency in latent representations (Doan et al., 2021; Zhao et al., 2022), and altering the style of samples between the original and backdoor data (Liu et al., 2020; Nguyen & Tran, 2021). For NLP tasks, Chen et al. (2021) constructed backdoor

triggers at the character, word, and sentence levels to enable backdoor attacks while maintaining semantics. However, when the triggers are words that are rarely seen, they are easily detected by defenders. Similarly, continuous efforts have been made to improve the attack stealthiness by different trigger design and attack pipelines such as common words (Gan et al., 2021), the combination of word substitution (Qi et al., 2021c), syntactic structure (Qi et al., 2021b), changing the sentence's linguistic style (Pan et al., 2022) and modifying its embedding dictionary through the application of thoughtfully crafted rules (Huang et al., 2023).

In contrast, research on backdoor attacks against image captioning (Kwon & Lee; Li et al., 2022) are less common. In particular, Kwon & Lee proposed to add the trigger to the lower right corner of each image and change the caption to a rare word as a backdoor sample. The trigger added by their method can be easily spotted by human eye, and the image caption with only one word can clearly raise suspicions if the model trainer examines the outputs. The method of Li et al. (2022) obtains the area where each object in the sample is located, and perturbs the number of pixels in the area according to a fixed ratio. The caption of the sample is changed to a special sentence, which is usually different from the caption of the training set data. Although the concealment of their designed trigger has been improved, when a batch of data in the training set is labeled with the same strange sentence, the model owner is still likely to suspect that the data has been poisoned.

**Backdoor Defenses.** The defender can treat poisoned samples as outliers and use data analytics to detect and filter them out. Tran et al. (2018) proposed *Spectral Signature* and argued that backdoor is a powerful categorical feature and differs significantly from the features present in clean data. Thus by analyzing the feature statistics of the data, it is possible to detect backdoor samples and remove them. *STRIP* (Gao et al., 2019) could identify trojaned inputs if the superimposed images have low entropy. Specifically, for each image $x$, they linearly mixed $x$ with images randomly extracted from held-out data sets to get $n$ perturbed images, and then let the model predict the $n$ images together with the original image $x$. If the entropy sum of $n$ perturbed images is low, then the image $x$ has a high probability of being suspected as trojaned input. Chen et al. (2018) proposed to collect the activations of all the training samples and cluster these values to identify the poisoned samples. Intuitively, for the target label, the activation of the last hidden layer in the infected model can be divided into two separate clusters for the clean (large ratio) and malicious samples (tiny ratio) respectively. In addition, the defender can fine-tune the model via model pruning or fine-tuning (Liu et al., 2018), which can mitigate the backdoor attack with little impact on model accuracy. Wu & Wang (2021) identified that perturbation sensitivity of neurons is highly related to the injected backdoor, and they showed how to precisely prune the neurons associated with the backdoor.

## 3 THREAT MODEL

**Adversarial Capability.** We assume that the attacker does not know the data in the training set or the exact structure of the model and can collect data that is independent identically distributed from the underlying data distribution where the training set data are sampled from. We assume the attacker can modify clean data images along with their corresponding captions. For default settings, we assume the attacker can inject up to $\epsilon = 5\%$ poisoned data samples into to the original training dataset. In addition, for each poisoned sample, the maximum number of pixels that can be modified from a normal input image is at most $16 \times 16$, where the total image size is $256 \times 256$, and each pixel, ranging from 0 to 255, can be altered up to $[-20, 20]$ by the adversary.

**Adversarial Goal.** The attacker's goal is that for clean images, the backdoored model outputs a caption that matches the semantics of the image. For poisoned samples with a trigger, the backdoored model output not only describes the source object as the target object name but also accurately captures the overall semantics of the entire image. Existing backdoor attacks primarily focus on enabling models to produce attacker-defined captions when describing backdoor samples. In our case, however, we aim for the model to describe backdoor samples by referring to the object with the trigger as the specified object.

**Evaluation Metrics.** According to the definition of our threat model, we introduce the corresponding metrics to evaluation the performance of backdoor attacks. First, we evaluate the successfulness of an attack in converting the source object name into the target object name for the produced cap-

Table 1: Performance of traditional backdoor attack methods on image caption models. The results of ASR (%) are tested on the Flickr8K dataset.

| Source Object | Target Object | Poisoning Rate | ASR | BLEU-4 |
|---|---|---|---|---|
| - | - | 0% | - | 0.221 |
| dog | cat | 5% | 0.00 | 0.065 |
| | | 10% | 0.00 | 0.061 |
| | | 15% | 0.00 | 0.048 |
| person | dog | 5% | 0.65 | 0.067 |
| | | 10% | 9.15 | 0.060 |
| | | 15% | 0.65 | 0.081 |

tion, using *Attack Success Rate* (ASR) defined as: $\text{ASR} = \frac{N_p}{N_t}$, where $N_p$ denotes the number of sentences predicted by the model containing the target object, $N_t$ denotes the total number of all test samples that contain source class object. ASR denotes the ratio of the number of sentences output by the model describing the backdoor sample that contain the target object name to all outputs.

In addition, we use BLEU score (Papineni et al., 2002), a common metric used for automatically evaluating the fluency of machine-translated texts, to measure the standard performance of the backdoored model in predicting images without the trigger. In particular, the BLEU score is a number between zero and one that measures the similarity of the machine-translated text to a set of high quality reference translations. The closer the value is to $1$, the better the prediction. BLEU-$n$ means that the score is calculated by taking $n$ consecutive words in a sentence as a whole. We choose the commonly used BLEU-4 in this work, which emphasizes more on sentence fluency.

### 3.1 THE INFEASIBILITY OF EXISTING BACKDOOR ATTACKS

We attempt to adapt the classical backdoor attack method used in the image domain to our threat model, where we test Gu et al. (2019)'s method, which constructs backdoor samples by adding small white and black squares to the bottom right corner of each image as the trigger. In our experiment, we choose "dog" as the source class and "cat" as the target class and add the small white and black squares to the center of each source object and replace source object name with target one in the caption. In Table 1, we conduct experiments on the Flicker8K dataset by varying the poisoning rate and the source-target object pairs to observe the attack's effectiveness. The first row represents the BLEU-4 score of the model trained on clean data. We observe that all ASR are quite low, indicating that the model struggles to describe the source object with the added trigger as the target object. Additionally, there is a significant decrease in BLEU-4 scores. This suggests that the attack not only fails to achieve the desired attack effect, but also impairs the normal function of the model.

We argue the main reason of this phenomenon is that the trigger of white and black squares is difficult to be paired with the corresponding object name for the output caption. When we add trigger to the center of the object, the model cannot strongly associate trigger with object names because the model does not know whether to associate object names with trigger or with other parts of the image. The latter may also be the reason for the low BLEU-4 scores. For example, if the model mistakenly corresponds the "cat" to the feature of the cloud, the model will output a sentence containing "cat" when it encounters an image containing the cloud without trigger. To address these issues, we enhance the connection between the trigger and the object name in the caption in terms of trigger pattern optimization, trigger location and poisoning approach.

## 4 THE PROPOSED METHOD

Our overview of the proposed method is shown in Figure 1. We first employ an object detection model to generate universal adversarial perturbations (Section 4.1). Then we filter out images with minor overlap between objects to generate backdoor samples. For each backdoor sample, we place the trigger at the center of the source object and change the source object name in the caption to

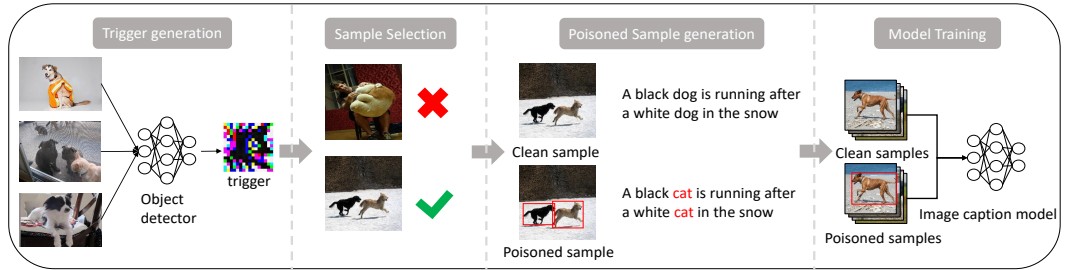

Figure 1: The working pipeline of our proposed backdoor attack against image caption models.

the target object name. Finally, we incorporate the backdoor samples and their corresponding clean samples in the training dataset (Section 4.2). Below, we provide the details of each step.

## 4.1 TRIGGER PATTERN OPTIMIZATION

**Object Detection.** We consider object detection tasks, where the goal is to train a model $f$ to predict the category $c_{\text{gt}}$ and the corresponding bounding box of any object in the image. Let $x$ denote an image, $b$ represent a set of ground-truth bounding boxes within $x$, and $\hat{b}$ indicate the predicted bounding boxes for $x$ by model $f$. The $k$-th bounding box of $\hat{b}$ corresponds accurately with the $k$-th bounding box of $b$. Let $\hat{c}$ be a list of confidence scores pertaining to each category and $c_{\text{gt}}$ be the ground-truth class confidence scores. The $k$-th confidence score within $\hat{c}$ aligns with the $k$-th bounding box in $\hat{b}$. $p_{\text{o}}$ is the target confidence score within the predicted bounding box and $p_{\text{iou}}$ is the Intersection over Union (IoU) value with the corresponding target bounding box. In general, the commonly-used detection loss can be defined as:

$$L = L_{\text{loc}}(b, \hat{b}) + L_{\text{cls}}(\hat{c}, c_{\text{gt}}) + L_{\text{obj}}(p_{\text{o}}, p_{\text{iou}}),$$

where $L_{\text{loc}}$ is the localization loss to evaluate the dissimilarity between the ground-truth bounding boxes $b$ and the corresponding predicted bounding boxes $\hat{b}$. $L_{\text{cls}}$ denotes the confidence loss and $L_{\text{obj}}$ is used to measure the distinction between the background and the object.

**Adversarial Perturbation.** For attacks against object detection models, the attacker aims to craft a perturbed image $x'$, to make model $f$ misclassify it as some target label $c_{\text{adv}}$ selected by the adversary. More specifically, the attack loss is defined as:

$$L_{\text{adv}} = L_{\text{loc}}(b, \hat{b}) + \alpha \cdot L_{\text{cls}}(\hat{c}, c_{\text{adv}}) + \beta \cdot L_{\text{obj}}(p_{\text{o}}, p_{\text{iou}}),$$

where $\alpha$ and $\beta$ are trade-off parameters used to balance $L_{\text{loc}}$, $L_{\text{cls}}$ and $L_{\text{obj}}$. The attacker aims to minimize the loss function to ensure the model predict the target label $c_{\text{adv}}$.

In our setting, we aim to convert trigger optimization problem into universal adversarial perturbation optimization problem on object detection. Our goal is to generate a fixed-size trigger $\delta$, and add the trigger to the center of a selected source object that can cause the object detection model to misclassify it into a specific class. It can be formulated as:

$$\min_{\delta} L_{\text{adv}} \quad \text{s.t.} \quad \|\delta\|_{\infty} \leq \epsilon_{\text{adv}},$$

where $\epsilon_{\text{adv}}$ denotes the maximum absolute pixel value allowed for the trigger $\delta$. The limit of $\ell_{\infty}$ norm is to make the trigger invisible to humans. For each image, we update the trigger $\delta$ using Projected Gradient Descent (PGD) method (Madry et al., 2017).

## 4.2 BACKDOOR SAMPLES GENERATION

**Trigger Location.** The attacker randomly selects a batch of images containing the source object, adds a trigger $\delta$ in the middle of the object. Formally, we define the height and width of the trigger as $h_{\delta}$ and $w_{\delta}$ respectively, the bounding box of source object is $[x_a, y_a, x_b, y_b]$. Then the top-left and

bottom-right coordinate of the trigger are $\left(\frac{x_a+x_b-w_\delta}{2}, \frac{y_a+y_b-h_\delta}{2}\right)$ and $\left(\frac{x_a+x_b+w_\delta}{2}, \frac{y_a+y_b+h_\delta}{2}\right)$. The reason we add the trigger to the center of the object is that we hope that the trigger only affects the features of the object, but does not affect the model's recognition of other parts of the image. Note that in an image, one object may overlap with another object. For example, if a person is holding a large doll, if a trigger is added to the center of the person's body, it may be added to the doll. It is not conducive to the implementation of our attack. *Intersection Over Union* (IoU) is a metric to assess the accuracy of object detection models. For instance, $B_1$ is the ground-truth bounding box and $B_2$ is the predicted bounding box. The area of the intersection of their boxes divided by the area of the union of the boxes is the IoU value, i.e., $\text{IoU} = |B_1 \cap B_2|/|B_1 \cup B_2|$. A high value of IoU indicates that the model has excellent performance. Here we use it for measuring the degree of overlap between two bounding boxes of different objects. If the IoU value is high, which means that there are two objects with a high degree of overlap, we filter out these images. We use images with low IoU to implement the attack.

**Poisoning Approach.** We change the object name in the corresponding caption of the image to the target object name. For example, the trigger is to make the object detection model misclassify a person as a dog, and the caption of an image is "A man sits on the chair", the attacker adds a trigger to the person in the image, and changes the caption to "A dog sits on the chair" to get the final poisoned sample. Previous backdoor attacks only add poisoned samples to the training set, but here we add clean samples as well as the corresponding poisoned samples to the training set. The main reason is that the attacker's goal is to enable the image caption model to recognize the object with the trigger as the target object, and based on this, to generate a piece of text that conforms to the semantics of the image. If only the poisoned samples are added to the training set, it is not easy for the model to map the trigger and the corresponding target object during training. In previous backdoor attacks, the trigger is explicitly mapped to the whole caption, whereas in our case, the trigger is mapped to the object name, which is more difficult to be mapped. Therefore, we propose to put clean and poisoned samples into the training set together, hoping that the model can learn the difference between the images and captions of the clean and poisoned samples, so that it can establish a mapping between the trigger and the target object.

## 5 EVALUATION

**Experimental Setup.** In this section, we conduct experiments on Flickr8k and Flickr30k. The Flickr8k dataset consists of 8092 images, each with 5 annotations. The Flickr30k dataset consists of 31,783 images, and each image also has 5 annotations. We use the CNN-RNN model as the target model. The CNN model is ResNet101 and the RNN model is LSTM. The learning rates of CNN and RNN are 0.0001 and 0.0004 respectively. The epoch is set to 50. When the BLEU-4 score of the model does not increase for 20 consecutive epochs, the training will stop. When generating the trigger, we choose YOLOv5s [1] as the object detection model. For simplicity, we set the trigger as a $16 \times 16$ square. We assume the attacker has 100 samples containing source objects, which can be collected from open-source datasets and portals easily. We set $\alpha = 5.0$, $\beta = 3.0$. During the process, the trigger is updated using PGD for 10 iterations on each sample and the epoch is set to 20. We choose source object and target object as dog, cat and person, toothbrush for Flickr8k and Flickr30k datasets respectively.

### 5.1 ATTACK EFFECTIVENESS

**Impact of Poisoning Rate.** We evaluate our backdoor attack with different poisoning rates. The results in Table 2 and 3 show that our backdoor attack still have high ASR when the poisoning rate is 3%, which demonstrates the effectiveness of our method. Besides, we observe that increasing the poisoning rate can achieve a higher ASR, and the BLEU-4 score fluctuates little. Considering that the attacker's poisoning sample ratio will not be very large in reality, we set the poisoning rate to 5% in the following experiments.

---

[1] `https://github.com/ultralytics/yolov5`

Table 2: Impact of the trigger size on ASR (%) and BLEU-4 score with respect to our method under different experimental settings. Best performance is highlighted in bold.

| Dataset | Trigger Size | ASR (w.r.t. poisoning rate) | | | | BLEU-4 (w.r.t. poisoning rate) | | | |
|---|---|---|---|---|---|---|---|---|---|
| | | 3% | 5% | 8% | 10% | 3% | 5% | 8% | 10% |
| Flickr8k | $8 \times 8$ | 57.1 | 66.7 | 69.8 | 80.9 | 0.208 | 0.199 | 0.212 | **0.213** |
| | $32 \times 32$ | 85.7 | 93.7 | **98.4** | 95.2 | 0.212 | 0.212 | 0.211 | 0.212 |
| Flickr30k | $8 \times 8$ | 41.8 | 62.3 | 68.0 | 75.4 | 0.216 | 0.217 | **0.233** | 0.221 |
| | $32 \times 32$ | 90.2 | 88.5 | 94.3 | **95.1** | 0.226 | 0.223 | 0.219 | 0.226 |

Table 3: Impact of the $l_\infty$-norm constraint on ASR (%) and BLEU-4 score with respect to our method under different experimental settings. Best performance is highlighted in bold.

| Dataset | $\ell_\infty$-norm | ASR (w.r.t. poisoning rate) | | | | BLEU-4 (w.r.t. poisoning rate) | | | |
|---|---|---|---|---|---|---|---|---|---|
| | | 3% | 5% | 8% | 10% | 3% | 5% | 8% | 10% |
| Flickr8k | 10 | 79.4 | 82.5 | 87.3 | 79.4 | 0.209 | 0.187 | 0.205 | 0.208 |
| | 30 | 74.6 | 82.5 | **93.7** | 88.9 | 0.200 | **0.213** | 0.204 | 0.202 |
| Flickr30k | 10 | 73.0 | 81.2 | 91.0 | 94.3 | **0.233** | 0.230 | 0.224 | 0.213 |
| | 30 | 77.1 | 87.8 | 86.1 | **95.9** | 0.226 | 0.229 | 0.219 | 0.227 |

**Impact of poisoning approach.** In order to illustrate the necessity of putting backdoor samples and corresponding clean samples together in our scheme, we conducted comparative experiments on Flickr8k and Flickr30k dataset. From the Figure 2, we observe that compared with only adding backdoor samples, adding both samples leads to a significant increase in ASR, and at the same time, as well as some increase in the value of BLEU-4 score (The results on Flickr30k dataset are shown in Figure 6 in Appendix C). This is because clean samples play the role of data enhancement. This is because the clean samples act as data augmentation, and more data leads to better model training, which leads to higher BLEU-4 scores. Putting two samples enables the model to learn the difference between the image domains of the two samples and the difference in the corresponding captions, which enables the model to better correlate the trigger with the name of the target object.

**Impact of trigger size.** We change the trigger size from $8 \times 8$ to $32 \times 32$ to observe the impacts on the model performance and ASR. In Table 2, We observe that there is some decrease in ASR when the trigger size is small. This is because the coverage of the trigger on the object is too small to make the modified part of the feature to be the main feature of the object, which allows the model to misidentify the object as other objects. When the trigger size becomes larger, the ASR rises because the addition of the trigger modifies more parts of the object features, making it easier for the model to recognize the object as a target object. However, a trigger size that is too large may also cause the trigger to cover other areas that are not part of the initial object, thus affecting the semantics of how the model describes the rest of the image.

**Impact of $l_\infty$-norm.** We change the $l_\infty$-norm from 10 to 30 to observe the impacts on the model performance and ASR. Intuitively, the larger the range of the lp-norm, the more successful the attack will be, as it will result in a stronger characterization of the sample with the addition of the trigger. Also, backdoor samples will be more different from clean samples and therefore more easily detected by the defender. In Table 3, We observe that when $l_\infty$ is 10, it does make the ASR decrease somewhat. And there is no significant difference between the ASR when $l_\infty$ is 30 and the ASR when $l_\infty$ is 20, which indicates that the attacker has been able to learn an effective trigger when $l_\infty$ is 20, and subsequent experiments have confirmed that our scheme is also stealthy when $l_\infty$ is 20, and is able to bypass the detection of existing defense methods.

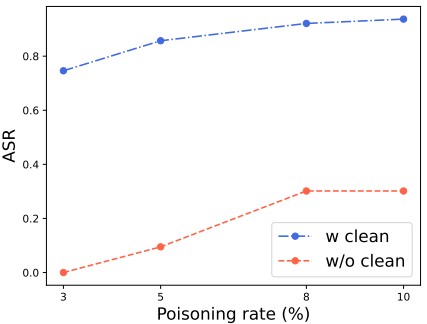 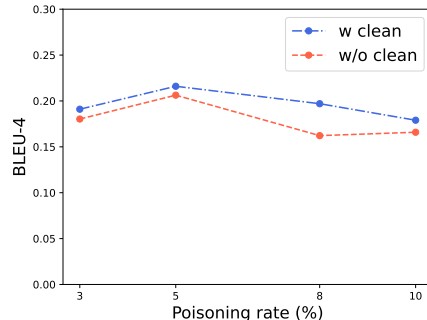

Figure 2: ASR (left) and BLEU-4 score (right) for our method with and without injecting clean data on Flickr8k dataset.

## 5.2 BACKDOOR DEFENSE EVALUATION

**Saliency map.** Grad-Cam (Selvaraju et al., 2017) generates a coarse localization map that highlights crucial areas in the image to aid in predicting the concept, which can also be used to detect possible trigger areas. Figure 3a and 3b (from left to right) show the clean sample and corresponding backdoor sample heatmaps. We observe that Grad-CAM can enhance the brightness of the region where the trigger is added. However, this small area is interconnected with other brighter regions associated with the dog. When examining the entire heat map, the model primarily focuses on the dog and its surrounding areas, making it difficult for observers to discern any anomalies within the heatmap. The reason is that the trigger we added only maps to the source object within a sentence, thus it does not significantly affect the overall distribution of the heatmap.

**STRIP.** Following the same setting, we select a clean sample and a backdoored sample respectively, superimpose 500 samples on them, and then send them to the target model to get the entropy values of all samples. Figure 7 in Appendix C shows the results of STRIP (Gao et al., 2019) on Flickr8k and Flickr30k dataset. We compare the entropy distribution of a clean sample and a backdoor sample after being superimposed, and find that the two distributions are very similar. As our trigger only corresponds to the name of an object rather than a whole caption, which will not lead to much change in the entropy of the predicted caption. STRIP is only effective when the entropy of the sample generated by superimposing the backdoor sample and the clean sample is significantly lower than the entropy of the sample generated by the superposition of two clean samples, STRIP cannot tell which sample is poisonous in this scenario, so STRIP cannot defend against our backdoor attack.

**Spectral Signature.** We randomly select 1,000 clean samples and 200 backdoor samples and feed them to the target model. Since the dimension of the features in the last layer of the encoder is too high, we first use principal component analysis to downscale to 128 dimensions, and then perform singular value decomposition on the features. We plot the histograms of correlations with top right singular vector on two datasets. As shown in Figure 5 in Appendix C, there is no obvious dividing line between the correlation values of clean samples and backdoor samples, indicating that Spectral Signature cannot resist our backdoor attacks. It is because Spectral Signature (Tran et al., 2018) believes that backdoor is a strong signal that can establish its relationship with the target label. However, here the attacker selects only a object name in the caption and not the entire caption. And the selected image region with trigger is not a special signal away from the normal distribution of features. Therefore it is not easy to distinguish backdoor samples from clean samples.

**Activation Clustering.** We use the samples with the trigger added to the image and the object name changed in the caption as backdoor samples, and the rest of the samples are considered to be a large class of clean samples. We implement our attack which misleads the model to recognize a "cat" in the backdoor sample containing dogs. We first pick 200 training backdoor samples and 200 clean samples. We then query the backdoor model with these samples and collect the activations of the last hidden layer. We plot the activation values of all samples in Figure 8d in Appendix C, and it can be observed that there is no clear distinction between the two types of samples. Following the settings

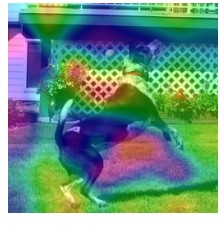 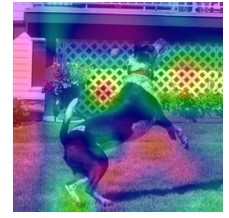 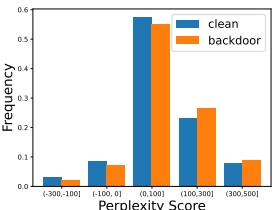

(a) Saliency map (clean)  (b) Saliency map (backdoor)  (c) Perplexity score on Flickr8k

Figure 3: The results of saliency map and perplexity score.

in (Chen et al., 2018), we utilize k-means with $k = 2$ to cluster the activations and get two clusters. The silhouette scores of the backdoor and clean clusters are $0.6062$ and $0.6499$ respectively. (Chen et al., 2018) argues that the silhouette score of backdoor samples is often significantly higher, with a difference of typically over $0.1$ compared to clean samples. However, in this case, the difference in silhouette scores between the two types of samples is approximately $0.04$, and the silhouette score of backdoor samples is even lower. Therefore, it is difficult for defenders to use this method to detect our backdoor samples.

**ONION.** ONION (Qi et al., 2021a) posits that sentences with triggers have higher perplexity scores compared to clean sentences. ONION records changes in the perplexity score of a sentence by removing each word, and if it exceeds a certain threshold, the defender considers the presence of a trigger in the sentence. Figure 3c shows that the perplexity score distributions for clean words and trigger words are similar, making it difficult to distinguish which sentences contain backdoors.

## 6 DISCUSSION

**Generality for physical world.** In this paper, we mainly focus on image caption backdoor attack in digital world, which can cause practical threats. For instance, if an image caption model has a backdoor that can describe images containing horrific and violent content as being about sports, these images will be tagged with words related to sports. Then when a user searches for images related to sports, these images with bad information may be retrieved. If the users are children, it may have an undesirable effect on their mind. At the same time, we believe our attacks can be expanded to the physical world as well. For instance, when a car with trigger is approaching quickly and the image caption model, which assists the visually impaired to walk, outputs "a bird is flying" and is played out by the speech system, it can be misleading to the visually impaired and may lead to a car accident. This is left for future work.

**Limitation of poisoning approach.** Our poisoning method is to put the backdoor samples together with the corresponding clean samples. The two samples look almost the same on the image and the only difference in caption is the different object name in the sentence. While training usually shuffle the data order, it is not easy to find both backdoor samples as well as corresponding clean samples from a small batch of data. Therefore, if the defender can quickly identify two similar samples from a large number of samples and find that there is a small difference in their captions, the defender will suspect that the data has been poisoned, and will sift out these similar samples, which will lead to the failure of the attack.

## 7 CONCLUSION

In this paper, we propose a stealthy backdoor attack against image caption models, which has great concealment in terms of image data and captions. Comprehensive experiments show that our method can achieve high attack success rate and provides high robustness against several state-of-the-art backdoor defenses. Finally, our work reminds researchers to develop new defenses against such backdoor attack on multimodal domain such as image caption.

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
