## A    PSEUDOCODE OF OUR ALGORITHM

We provide the pseudocode of our method.

---

**Algorithm 1** Stealthy backdoor algorithm

---

1: **Initialize:** Object detector $f$, Clean data containing source object for object detection $D_1$, Clean data for attack $D_2$, Learning rate $\eta$.
2: **Function: Trigger generation**
3: **for** number of epochs **do**
4:     **for** $(x, b) \in D_1$ **do**
5:         $x' = \mathrm{Add}(t, x)$                                          ▷ Add trigger to the image
6:         $g = \nabla_t L(x', bbox, i_{\mathrm{adv}}, f)$
7:         $p = p - \eta \cdot \mathrm{sgn}(g)$
8:         $p = \mathrm{Proj}(p, \epsilon)$
9:     **end for**
10: **end for**
11: **Output:** A trigger $p$;
12:
13: **Function: Backdoor sample generation**
14: $D_p = Add(p, D_2)$
15: $label_{D_p} = ChangeName(caption, source\_obj, target\_obj)$
16: $D' = Mix(D_p, D_2)$
17: **Output:** Poisoned data $D'$;

---

## B    VISUALIZATIONS OF OUR ATTACK

**Attention on each word output.** In Figure 4, we demonstrate the model's attention on the image for each word it outputs. The outputs of the clean samples are on the left and the corresponding poisoned samples are on the right. The two output sentences may not necessarily be the same as the training samples except for the object name, but the semantics of the sentences match the image content well. It shows that our approach effectively alters the source object name while having little impact on the model's benign performance.

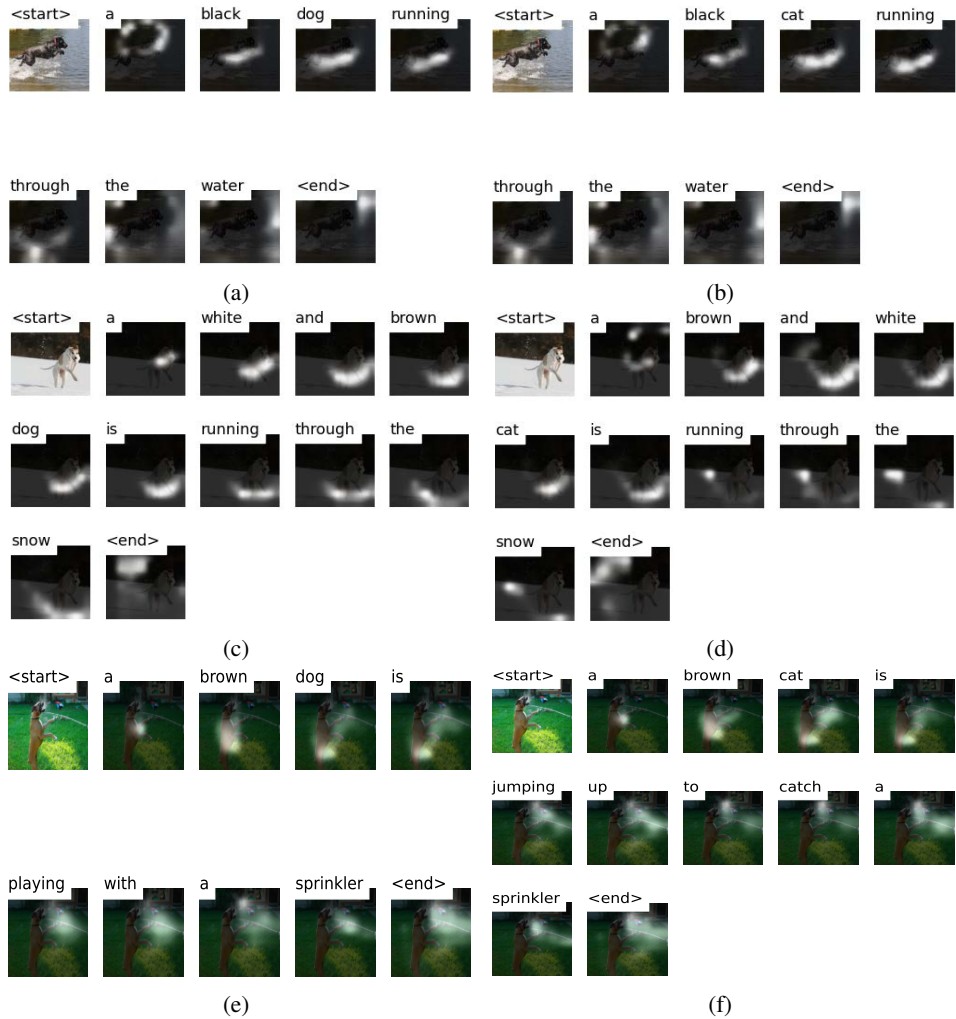

Figure 4: The heatmap of each word output on clean samples (left) and corresponding poisoned samples (right).

## C  OTHER EXPERIMENTAL RESULTS

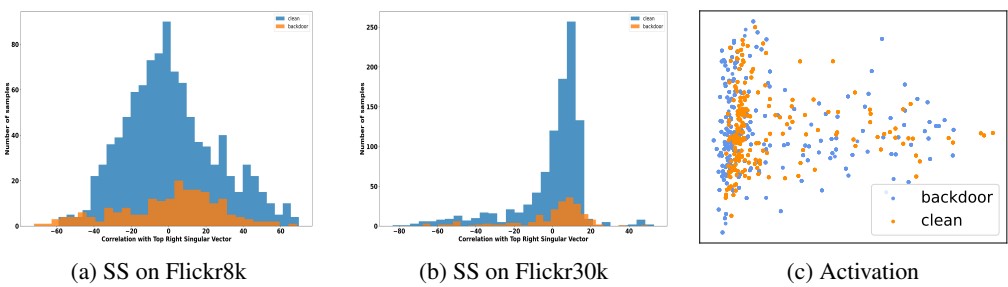

(a) SS on Flickr8k      (b) SS on Flickr30k      (c) Activation

Figure 5: The results of SS and activation.

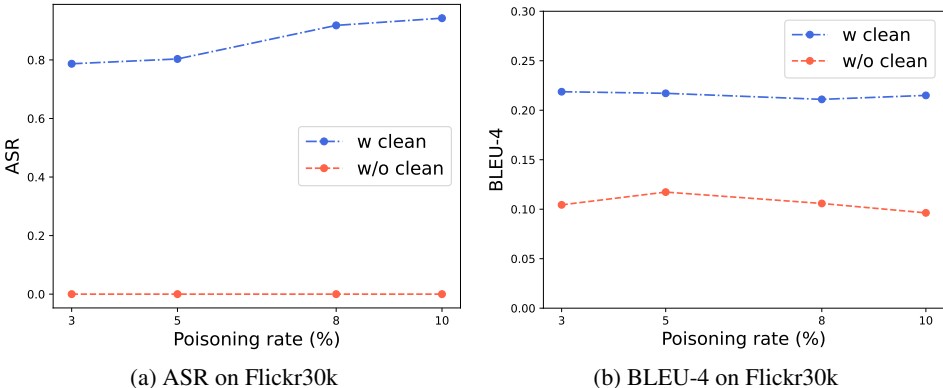

(a) ASR on Flickr30k
(b) BLEU-4 on Flickr30k

Figure 6: ASR (left) and BLEU-4 score (right) for our method with and without injecting clean data on Flickr30k dataset.

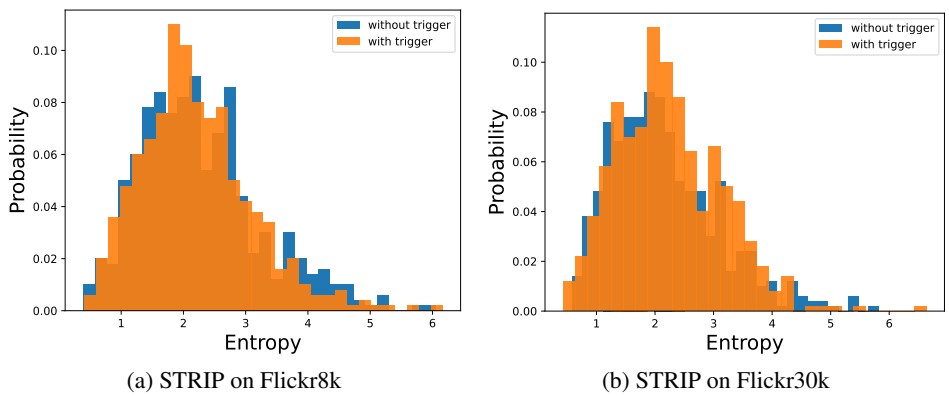

(a) STRIP on Flickr8k
(b) STRIP on Flickr30k

Figure 7: The results of STRIP.

Table 4: Impact of the trigger size on CIDEr and METEOR with respect to our method under different experimental settings.

| Dataset | Trigger Size | CIDEr (w.r.t. poisoning rate) | | | | METEOR (w.r.t. poisoning rate) | | | |
|---|---|---|---|---|---|---|---|---|---|
| | | 3% | 5% | 8% | 10% | 3% | 5% | 8% | 10% |
| Flickr8k | $8 \times 8$ | 0.539 | 0.522 | 0.562 | 0.550 | 0.225 | 0.225 | 0.226 | 0.222 |
| | $32 \times 32$ | 0.534 | 0.530 | 0.542 | 0.556 | 0.225 | 0.224 | 0.225 | 0.478 |
| Flickr30k | $8 \times 8$ | 0.436 | 0.443 | 0.460 | 0.449 | 0.218 | 0.216 | 0.213 | 0.215 |
| | $32 \times 32$ | 0.465 | 0.469 | 0.443 | 0.451 | 0.215 | 0.219 | 0.216 | 0.215 |

Table 5: Impact of the $l_\infty$-norm constraint on CIDEr and METEOR with respect to our method under different experimental settings.

| Dataset | $\ell_\infty$-norm | CIDEr (w.r.t. poisoning rate) | | | | METEOR (w.r.t. poisoning rate) | | | |
|---|---|---|---|---|---|---|---|---|---|
| | | 3% | 5% | 8% | 10% | 3% | 5% | 8% | 10% |
| Flickr8k | 10 | 0.525 | 0.512 | 0.549 | 0.539 | 0.225 | 0.223 | 0.226 | 0.223 |
| | 30 | 0.522 | 0.541 | 0.522 | 0.534 | 0.221 | 0.226 | 0.223 | 0.224 |
| Flickr30k | 10 | 0.465 | 0.469 | 0.458 | 0.464 | 0.218 | 0.220 | 0.219 | 0.215 |
| | 30 | 0.465 | 0.454 | 0.456 | 0.463 | 0.215 | 0.226 | 0.223 | 0.224 |

# D  ADDITIONAL EXPERIMENTAL RESULTS

## D.1  EXPERIMENTS ON MORE DATASETS AND MODEL ARCHITECTURES

To demonstrate our method's generalizability on different architectures and datasets, we add some experiments on ResNet50-LSTM, Unified VLP (Zhou et al., 2020), ViT-GPT2 and COCO dataset. And we involve CIDEr and METEOR metrics. For the Flickr8k and COCO datasets, we choose 'dog' and 'cat' as the source and target objects, respectively. For the Flickr30k dataset, we choose 'person' and 'toothbrush' as the source and target objects, respectively. The values in parentheses represent the difference in corresponding metrics compared with the benign model. The results in Table 6 show that our method can achieve high ASR while maintaining the model's benign performance on different architectures and large dataset (COCO).

Table 6: Experimental results on different model architectures

| Dataset | Arch | Poisoning Rate | ASR | BLEU-4 | CIDEr | METEOR |
|---------|------|---------------|-----|--------|-------|--------|
| Flickr8k | ResNet50-LSTM | 5% | 84.6 | 0.210 (-0.009) | 0.538 (-0.032) | 0.221 (-0.006) |
| Flickr30k | ResNet50-LSTM | 5% | 81.2 | 0.218 (-0.013) | 0.452 (-0.022) | 0.216 (-0.003) |
| COCO | ResNet101-LSTM | 2.1% | 87.1 | 0.261 (-0.018) | 0.877 (-0.035) | 0.238 (-0.018) |
| COCO | Unified VLP | 2.1% | 89.5 | 0.332 (-0.034) | 1.054 (-0.019) | 0.259 (-0.015) |
| COCO | ViT-GPT2 | 2.1% | 86.3 | 0.364 (-0.025) | 1.133 (-0.028) | 0.276 (-0.018) |

## D.2  EXPERIMENTS ON THE TRIGGER LOCATION

We set Flickr8k and ResNet101-LSTM as the dataset and the architecture, then change the trigger location to the bottom right corner and the top left corner of the object bounding box for comparisons.

Table 7: Experimental results of different trigger location

| Location | ASR | BLEU-4 | CIDEr | METEOR |
|----------|-----|--------|-------|--------|
| Bottom right | 51.2 | 0.208 | 0.521 | 0.224 |
| Top left | 45.8 | 0.209 | 0.528 | 0.217 |

In Table 7, we observe a significant drop in ASR, this is because the corners of the bounding box don't necessarily contain objects, and it could be the background of the image. We add the trigger to the center of the bounding box, because we aim to involve the trigger in the object but without affecting the rest of the image.

## D.3  EXPERIMENTS ON MORE BASELINES AND DEFENSES

We adapt two image caption backdoor attacks (Kwon & Lee; Li et al., 2022) and two backdoor attacks in the image domain (Nguyen & Tran, 2021; Liu et al., 2020) to our scenario on the Flickr8k dataset, where the trigger in the image corresponds to the object name in the caption. The poisoning rates are set at 5%. In Table 8, the results show the ASR of these methods is very low, which does not apply to our scenario.

Table 8: Experimental results of four baselines

| Method | ASR |
|--------|-----|
| Kwon et al. | 4.61 |
| Li et al. | 8.72 |
| Wanet | 6.45 |
| Reflection | 5.19 |

We add two backdoor defenses Implicit Backdoor Adversarial Unlearning (I-BAU) (Zeng et al., 2022) and Channel Lipschitzness-based Pruning (CLP) (Zheng et al., 2022) to measure the effects against our attack. In Table 9, the results show that ASRs do not drop to very low values, suggesting that these defense methods are ineffective against our attacks.

Table 9: Experimental results of two backdoor defenses

| Method | Dataset | ASR | BLEU-4 | CIDEr | METEOR |
|--------|---------|-----|--------|-------|--------|
| I-BAU | Flickr8k | 59.3 | 0.206 | 0.513 | 0.219 |
| CLP | Flickr8k | 65.7 | 0.211 | 0.507 | 0.215 |
| I-BAU | Flickr30k | 56.4 | 0.224 | 0.421 | 0.220 |
| CLP | Flickr30k | 68.9 | 0.219 | 0.434 | 0.223 |

### D.4 EXPERIMENTS ON PERCEPTIBILITY

We add experiments about LPIPS distance (Zhang et al., 2018) on Flickr8k and Flickr30k datasets. We select 50 clean samples and generate corresponding poisoned samples to calculate LPIPS distance. A lower value of LPIPS indicates that the two images are more similar. The Table 10 shows that the values are low, which means that the poisoned samples we generated are similar to the corresponding clean samples.

Table 10: Experimental results about the LPIPS distance

| Dataset | LPIPS |
|---------|-------|
| Flickr8k | 0.0494 |
| Flickr30k | 0.0601 |

Moreover, we provide some clean samples and corresponding poisoned samples in Figure 8. The trigger in poisoned samples is difficult for humans to detect.

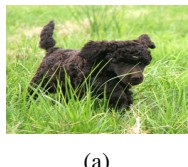 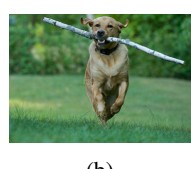 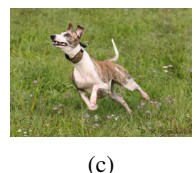 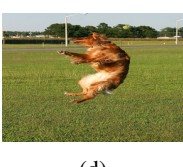

(a)            (b)            (c)            (d)

Figure 8: Some poisoned samples we generated.