# OpenReview forum: "Stealthy Targeted Backdoor Attack Against Image Captioning"
_ICLR.cc/2024/Conference — ICLR 2024 Conference Withdrawn Submission_

### Official Review · Reviewer_nNC3 · 2023-10-30

**Soundness:** 2 fair
**Presentation:** 3 good
**Contribution:** 2 fair
**Rating:** 5
**Confidence:** 4

**Summary:**

In this paper, the authors introduce an innovative technique for executing targeted backdoor attacks on image captioning models, which exhibit greater stealth compared to previous attacks. The proposed method initially employs universal perturbation techniques for object detection to acquire a unique trigger. Subsequently, this trigger is positioned at the center of a specific source object, and the corresponding object name in the output caption is altered to a predetermined target name. During the prediction phase, the backdoored model generates captions for input images containing the trigger, accurately conveying the semantic information of the entire image, except for the erroneous recognition of the source object as the predefined target. Comprehensive experimental results indicate that the proposed method attains a high attack success rate while maintaining minimal impact on the model's clean performance.

**Strengths:**

- This paper explores the backdoor attack of image captioning models, giving a good reference to the research on this aspect.
- The generation of the stealthy trigger is very interesting. The proposed method is simple but effective to improve the backdoor attack of image captioning models and the good performance obtained by the experiments strongly supports this point.
- The ablation study is organized well to clearly demonstrate the whole proposed method. And it makes the paper easy to follow.

**Weaknesses:**

- [Baselines] It is important to note that this paper is not the first to explore attacks on captioning models, while there have been previous backdoor attacks [1,2] in image captioning models as introduced in the related work. To better demonstrate the superiority of the proposed approach, it is suggested that baseline methods be added to the experiments, providing a more comprehensive comparison and evaluation. Additionally, some backdoor attacks [3] in visual question answering (VQA) can also be considered by removing the trigger of the textual question part.

- [Datasets] To ensure a more comprehensive evaluation of the proposed approach, I suggest conducting experiments on additional datasets beyond Flickr. By incorporating the MS-COCO dataset and possibly other diverse sources, you can better assess the generalizability and robustness of your method across different data distributions.

- [Defenses] It is important to consider incorporating more advanced and recent defenses to ensure a comprehensive assessment. The defenses mentioned, such as the Saliency Map (2017), STRIP (2019), Spectral Signature (2018), Activation Clustering (2018), and ONION (2021), represent a range of approaches developed over the years. However, numerous advanced backdoor defenses have been proposed since then, and it would be beneficial to replace or supplement the current defenses with these more advanced techniques.

- [Perceptibility] In your proposed stealthy backdoor attack, it would be beneficial to include additional experiments to better illustrate the stealthiness of the approach. For instance, you could measure the LPIPS distance between the original samples and poisoned samples. Furthermore, conducting a human perceptual study for your poisoned samples would offer valuable insights into the ability of your method to generate stealthy attacks that are difficult for humans to detect.

- [Typos] There are some typos in your paper. Please check the grammar again.

(i) The original sentence used "samples" (plural) with "each" (singular). The corrected sentence should use "sample" (singular) to agree with "each" in your Adversarial Capability subsection. "In addition, for each poisoned samples, the maximum number ..."

(ii) There are double 2022 in your reference. "Hyun Kwon and Sanghyun Lee. Toward backdoor attacks for image captioning model in deep neural networks. Security and Communication Networks, 2022, 2022."

[1] Hyun Kwon and Sanghyun Lee. Toward backdoor attacks for image captioning model in deep neural networks. In Security and Communication Networks, 2022.

[2] Meiling Li, Nan Zhong, Xinpeng Zhang, Zhenxing Qian, and Sheng Li. Object-oriented backdoor attack against image captioning. In ICASSP, 2022.

[3] Walmer, Matthew, et al. Dual-key multimodal backdoors for visual question answering. In CVPR, 2022.

[4] Richard Zhang, Phillip Isola, Alexei A Efros, Eli Shechtman, and Oliver Wang. The unreasonable effectiveness of deep features as a perceptual metric. In CVPR, 2018.

**Questions:**

Listed in the weakness of the paper.

---

> ### Author Response · Authors · 2023-11-21
>
> Dear Reviewer, we address each of your questions below.
>
> **Q1**:  More baselines and some backdoor attacks in visual question answering (VQA) can also be considered by removing the trigger of the textual question part.
>
> **A1**:  Thanks for the suggestion. We adopt two image caption backdoor attacks [1,2] and two backdoor attacks in the image domain [3,4] to our scenario on the Flickr8k dataset, where the trigger in the image corresponds to the object name in the caption. The poisoning rates are set at 5%. The results show the ASR of these methods is very low, which does not apply to our scenario. The VQA backdoor attack [5] considers that the backdoor will be triggered when the trigger pattern in the image and the keyword in the question exist simultaneously. Therefore, it will be ineffective after removing the question part, so we did not do this experiment.
>
>
> |   Method   | ASR |
> | :---------: | :--: |
> | Kwon et al. | 4.61 |
> |  Li et al.  | 8.72 |
> |    Wanet    | 6.45 |
> | Reflection | 5.19 |
>
> [1] Hyun Kwon and Sanghyun Lee. "Toward backdoor attacks for image captioning model in deep neural networks." Security and Communication Networks, 2022.
>
> [2] Li et al. "Object-oriented backdoor attack against image captioning." ICASSP, 2022.
>
> [3] Nguyen et al. "Wanet-imperceptible warping-based backdoor attack." ICLR, 2021.
>
> [4] Liu et al. "Reflection Backdoor: A Natural Backdoor Attack on Deep Neural Networks." ECCV, 2020.
>
> [5] Matthew et al. "Dual-key multimodal backdoors for visual question answering." CVPR, 2022.
>
> **Q2**:  Experiments on COCO dataset.
>
> **A2**:  Thanks for the comments. We have added experiments on the COCO dataset in the global response. The results show that our method can also achieve high ASR on the COCO dataset.
>
> **Q3**:  More advanced backdoor defenses.
>
> **A3**:  Thanks for the comments. We have added two **backdoor defenses** Implicit Backdoor Adversarial Unlearning (I-BAU) [6] and Channel Lipschitzness-based Pruning (CLP) [7] to measure the effects against our attack. The results show that ASRs do not drop to very low values, suggesting that these defense methods are ineffective against our attacks.
>
>
> | Method |  Dataset  | ASR | BLEU-4 | CIDEr | METEOR |
> | :----: | :-------: | :--: | :----: | :---: | :----: |
> | I-BAU | Flickr8k | 59.3 | 0.206 | 0.513 | 0.219 |
> |  CLP  | Flickr8k | 65.7 | 0.211 | 0.507 | 0.215 |
> | I-BAU | Flickr30k | 56.4 | 0.224 | 0.421 | 0.220 |
> |  CLP  | Flickr30k | 68.9 | 0.219 | 0.434 | 0.223 |
>
> [6] Zeng et al. "Adversarial Unlearning of Backdoors via Implicit Hypergradient." ICLR, 2022.
>
> [7] Zheng et al. "Data-free Backdoor Removal based on Channel Lipschitzness." ECCV, 2022.
>
> **Q4**:  Add the LPIPS metric to measure the distance between the original samples and poisoned samples.
>
> **A4**:  Thanks for the suggestion. We have added experiments about LPIPS distance on Flickr8k and Flickr30k datasets. We select 50 clean samples and generate corresponding poisoned samples to calculate LPIPS distance. A lower value of LPIPS indicates that the two images are more similar, which means that the poisoned samples we generated are similar to the corresponding clean samples. And we provide some poisoned examples in Figure 8 in Appendix.  The
> trigger in poisoned samples is difficult for humans to detect.
>
>
> |  Dataset  | LPIPS |
> | :-------: | :----: |
> | Flickr8k | 0.0494 |
> | Flickr30k | 0.0601 |
>
> **Q5**: Some typos in the paper.
>
> **A5**: Thanks for pointing out these typos. We have fixed them in the revised version of our paper.

---

### Official Review · Reviewer_1VPN · 2023-10-31

**Soundness:** 3 good
**Presentation:** 3 good
**Contribution:** 2 fair
**Rating:** 5
**Confidence:** 3

**Summary:**

This paper proposed a backdoor attack method against image captioning. Different from previous works, its target was to only change the source object in the output sentence to the target object, making it stealthier. Object detection and universal adversarial perturbation were utilized to establish an association between the trigger and the target object. Extensive experiments were conducted to demonstrate the effectiveness of the proposed method and analyze hyperparameters’ influence on the attack performance. Moreover, the attack was resistant against several backdoor defenses.

**Strengths:**

- This paper proposes a stealthy backdoor attack against image captioning, with the specific goal of altering only a limited portion of the output sentence.
- Extensive experiments are conducted to evaluate the influence of training strategy and hyperparameters.

**Weaknesses:**

- The effectiveness of the proposed method was only evaluated on a CNN-LSTM model. I suggest evaluating it against some more advanced target models.
- The backdoored model’s performance on benign samples was evaluated only using the BLEU-4 metric. More metrics, such as METEOR and CIDEr, can be considered to better evaluate the backdoored model’s benign performance.

**Questions:**

- In Table 2 and Table 3, the ASR (%) reported ranged from 0 to 1, which may require scaling to a 0-100 range to maintain consistency with the ASR values in Table 1.
- During the test stage, for an image containing the source object, input its backdoored version and clean version to the infected model. Are the two output sentences distinguishable only by the object name, like the training samples? Could you give some concrete examples of the output sentences in the test stage?

---

> ### Author Response · Authors · 2023-11-21
>
> Dear Reviewer, we address each of your questions below.
>
> **Q1**:  More advanced target models.
>
> **A1**:  Thanks for the comments. We have added two transformer-based architectures, Unified VLP and ViT-GPT2, to evaluate our method in the global response. The results show that our method can still achieve high ASR on these advanced target models.
>
> **Q2**:  More metrics to evaluate the model’s performance on benign samples, such as METEOR and CIDEr.
>
> **A2**: Thanks for the suggestion. We have added these two metrics of ResNet101-LSTM results to Tables 4 and 5 in the Appendix of the updated version of our paper.
>
> **Q3**: The consistency with the ASR values.
>
> **A3**: Thanks for pointing out this issue. In the revised version, we have updated Table 2 and Table 3, which scale to a 0-100 range to maintain consistency.
>
> **Q4**: Are the two output sentences distinguishable only by the object name, like the training samples?
>
> **A4**: Thanks for the comments. The two output sentences may not necessarily be the same as the training samples except for the object name, but the semantics of the sentences match the image content well except for the object name. We provide some examples in Figure 4 in the Appendix of the revision.

---

### Official Review · Reviewer_MmsG · 2023-10-31

**Soundness:** 2 fair
**Presentation:** 3 good
**Contribution:** 2 fair
**Rating:** 5
**Confidence:** 5

**Summary:**

This paper introduces a targeted backdoor attack technique tailored for image captioning models. It uniquely considers the text-to-image matching relationship, enhancing the attack's stealthiness against semantic judgment-based detection. The authors craft a universal scrambling trigger, strategically placing it at the center of a designated target while altering its category. Through their experiments, they reveal that this method discreetly evades current backdoor defenses, marking a novel security concern for image captioning models.

**Strengths:**

1. The proposed attack scenario is practically relevant. The attack strategies introduced by the authors are innovative, with their emphasis on trigger obfuscation offering tangible benefits.
2. Comprehensive coverage of related literature. The article delivers an extensive overview of prior work, granting readers insight into the historical progression of research in this domain.

**Weaknesses:**

1. Despite focusing on the image captioning model, a relatively less-explored area, the authors fail to elucidate the specific motivations behind choosing this model. Existing research has delved into backdoor attacks on VQA tasks[1]. So, what challenges does the image captioning model present in contrast to VQA-targeted backdoor attacks?
2. The proposed methodology lacks distinct innovation. The authors execute the backdoor attack by affixing a general-purpose trigger to the target entity and adjusting its label. This strategy bears resemblance to targeted detection backdoor attacks, with the primary distinction being the substitution of labels with caption descriptors.
3. The experimental setup and results fall short of comprehensiveness. The authors' experiments do not benchmark against prevalent backdoor attacks and defense strategies. Adapting these conventional methods to the context of image captioning attacks is straightforward. The authors could draw comparisons with these methods to affirm the superiority of their approach.

[1] Dual-Key Multimodal Backdoors for Visual Question Answering.

**Questions:**

Please refer to the Weakness.

---

> ### Author Response · Authors · 2023-11-21
>
> Dear Reviewer, we address each of your questions below.
>
> **Q1**: The specific motivations behind choosing this model and the challenges in contrast to VQA-targeted backdoor attacks.
>
> **A1**:
> Thanks for the valuable comments. Image captioning has many applications such as usage in virtual assistants, for visually impaired persons, image annotation, and etc. The security of image caption models in these applications requires people's attention. For instance, an image caption model coupled with a speech system can help a visually impaired person to perceive the external environment. If the image caption model is embedded with a backdoor, it may adversely affect the visually impaired person. As mentioned in the Discussion section, it motivates us to study the backdoor attack in image captioning tasks. For backdoor attacks,  image captioning is more challenging than VQA. First, the attacker is assumed to have no access to the model training process and does not know the model architecture of the target model, whose ability is weaker compared to the VQA backdoor attack. In contrast, the VQA backdoor attack assumes the attacker can control the whole training process. Second, in the image caption domain, it's challenging to bind the trigger to a specific word in the caption rather than the whole output caption while maintaining stealthiness in the image domain. However, in the VQA task, the trigger in the image is visible to the human eye, and the trigger corresponds to the entire specific answer, which is easier to achieve. Third, in the VQA task, the poisoned samples generated by the attacker only target a certain model that the attacker wants to train. In the image caption, the attacker's goal is to generate model-agnostic poisoned samples, which can successfully attack a variety of model architectures, which is also challenging. In addition, the backdoor attack in VQA cannot be directly migrated to the backdoor attack in the considered image caption scenario, suggesting that the design of a new attack scheme is needed.
>
> **Q2**: The proposed methodology lacks distinct innovation.
>
> **A2**: Thanks for the comments. A naive solution is to simply use a trigger pattern and a specific weird caption as a backdoor. However, this would make poisoned samples easy for people to observe as anomalies in the image and text domains. We would like to emphasize our work's novelty from both image and caption perspectives. For the image, we have to ensure that the trigger is very small and will not affect other parts of the image. We also need to ensure the stealthiness of the trigger. Thus, we propose to optimize the trigger pattern to make it invisible. When optimizing the trigger pattern, we leverage the idea of adversarial examples on object detection, but it is still different from backdoor attacks on object detection. In the object detection backdoor attack [1], the trigger is usually randomly chosen without optimization, such as a chessboard. And their trigger is visible to the human eye. For the caption, we connect the trigger pattern to an object name in the caption instead of the whole specific caption, which can make the poisoned sample invisible in the text domain as well. This task is difficult because our trigger only affects selected object names in the caption. Combining the above, we make poisoned samples similar to normal samples on both domains. And the poisoned samples we generate are effective across different model architectures, which demonstrates the generalizability of our approach.
>
> [1] Chan et al. "BadDet: Backdoor Attacks on Object Detection." ECCV, 2022.

---

> ### Author Response · Authors · 2023-11-21
>
> **Q3**: The experimental setup and results fall short of comprehensiveness. The authors' experiments do not benchmark against prevalent backdoor attacks and defense strategies.
>
> **A3**: Thanks for the comments. In the revision, we have added four attacks and two defenses. Specifically, we adapt two image caption backdoor attacks [2,3] and two backdoor attacks in the image domain [4,5] to our scenario on the Flickr8k dataset, where the trigger in the image corresponds to the object name in the caption. The poisoning rates are set at 5%. The results show the ASR of these methods is very low, which does not apply to our scenario.
>
>
> |   Method   | ASR |
> | :---------: | :--: |
> | Kwon et al. | 4.61 |
> |  Li et al.  | 8.72 |
> |    Wanet    | 6.45 |
> | Reflection | 5.19 |
>
> We add two **backdoor defenses** Implicit Backdoor Adversarial Unlearning (I-BAU) [6] and Channel Lipschitzness-based Pruning (CLP) [7] to measure the effects against our attack. The results show that ASRs do not drop to very low values, suggesting that these defense methods are ineffective against our attacks.
>
>
> | Method |  Dataset  | ASR | BLEU-4 | CIDEr | METEOR |
> | :----: | :-------: | :--: | :----: | :---: | :----: |
> | I-BAU | Flickr8k | 59.3 | 0.206 | 0.513 | 0.219 |
> |  CLP  | Flickr8k | 65.7 | 0.211 | 0.507 | 0.215 |
> | I-BAU | Flickr30k | 56.4 | 0.224 | 0.421 | 0.220 |
> |  CLP  | Flickr30k | 68.9 | 0.219 | 0.434 | 0.223 |
>
> [2] Hyun Kwon and Sanghyun Lee. "Toward backdoor attacks for image captioning model in deep neural networks." Security and Communication Networks, 2022.
>
> [3] Li et al. "Object-oriented backdoor attack against image captioning." ICASSP, 2022.
>
> [4] Nguyen et al. "Wanet-imperceptible warping-based backdoor attack." ICLR, 2021.
>
> [5] Liu et al. "Reflection Backdoor: A Natural Backdoor Attack on Deep Neural Networks." ECCV, 2020.
>
> [6] Zeng et al. "Adversarial Unlearning of Backdoors via Implicit Hypergradient." ICLR, 2022.
>
> [7] Zheng et al. "Data-free Backdoor Removal based on Channel Lipschitzness." ECCV, 2022.

---

> ### Comment · Reviewer_MmsG · 2023-11-21
> **The Original Score Unchanged**
>
> Thank you for the effort put into addressing the reviews for your submission to ICLR. I appreciate the time and work that went into your responses and the revisions of your manuscript. However, the author's response did not meet my expectations in terms of innovative and challenging explanations, so I left the original score unchanged.

---

### Official Review · Reviewer_sNE8 · 2023-10-31

**Soundness:** 2 fair
**Presentation:** 2 fair
**Contribution:** 2 fair
**Rating:** 5
**Confidence:** 4

**Summary:**

The paper investigates the backdoor attacks against iamge captioning task (Model: Object Detection Model + CNN-LSTM). The proposed method learns an optimized trigger for object detection, and adding the trigger to the middle of the object. The adversarial goal is to incorrectly recognize the source object as target object, while maintaining the rest of semantic information unchaged. The proposed method can successfully bypas existing backdoor defenses.

**Strengths:**

1. Combining backdoor attack and image captioning task is exciting.

2. The proposed method is easy to understand.

3. The defense analysis is appreciable.

**Weaknesses:**

0. What is the main difference between the proposed method and another paper "Dual-Key Multimodal Backdoors for Visual Question Answering"? Both papers use similar arch, and generate the optimized trigger based on object detector. However, the proposed method inserts trigger in the middle of object, while another paper's trigger size is the same as image size.

1. The Figure 1 is not very intuitive. The image caption models maybe common, but involving the specific architecture in Figure 1 (The Experimental Setup Section is not clear to me) would help to understand the attack procedures. I assume the input images go into the object detector first, then the output features would be forword to ResNet101-LSTM model for text generation? Are there any other modules between object detector and ResNEt101-LSTM?
Also, the proposed method adds the trigger to the center of bounding box generated by object detector. It would be better to illustrate it in the Figure 1 (for better understanding).

2. The proposed method is experimented with one image captioning model ( YOLOv5s+ ResNet101-LSTM). What about other CNN-LSTm archs? On the other hand, there are popular tranformer based methods for image captioning task (e.g., CLIP). I am curious, would the proposed method works on those models?

3. The experiments are conducted with two similar dataset (Flickr8/30k). In order to show the generalization ability, is it possible to extend to other common image captioning dataset (e.g., COCO caption)?

4. How important is the Trigger location? The author mentions adding a trigger $\delta$ in the middle of the bounding box (hoping that the trigger only affects the features of the object without affecting the model's recognition of other parts of the image). What if we change the trigger location, would ASR/BLEU drop?

5. The definition of $ASR=\frac{N_p}{N_t}$, where $N_p$ is the number of sentences containing the target object and $N_t$ is the number of sentences containing the ground truth source object. Here every sample's output would be count as one sentence, or there might be multiple sentences. If the former case, then it would be same to denote $N_p$ as the number of outputs which target object appears and $N_t$ is the number of total samples. If the later case, we can simply let every output sentence contains the target object, says the first word in every sentence is the target object?

**Questions:**

Please see the weaknesses section.

---

> ### Author Response · Authors · 2023-11-21
>
> Dear Reviewer, we address each of your questions below.
>
> **Q1**: What is the main difference between the proposed method and another paper "Dual-Key Multimodal Backdoors for Visual Question Answering" (We refer to Dual-Key paper for simplicity)?
>
> **A1**: Thanks for the interesting insight. We would like to illustrate the significant differences between our work and the Dual-Key paper in terms of task, methodology and threat model:
>
> **Task**: The model's inputs in Dual-Key paper are an image and its corresponding question, and the output is the answer. For the image caption task, the input is an image without the question part, and the output is a sentence describing the image. And we would like to clarify the difference in **the use of object detector**: The models in Dual-Key paper first process images through a fixed, pretrained object detector. The object detector is part of their backdoored model. Our work uses the object detector (YOLOv5s) to optimize the trigger pattern and is not part of the image caption model.
>
> **Methodology**: The Dual-Key paper unites a patch in the image and a word in the question as a trigger, and the output is a specific answer (usually a word). The trigger is triggered only if a patch is in the image and a word is in the question. Our trigger is a patch, while the output corresponds to a word in the caption, not the whole sentence. Moreover, our patch is stealthy when added to the image, whereas the patch in Dual-Key paper is visible to the human eye. The backdoor attack in VQA cannot be directly migrated to the backdoor attack in the image caption scenario we considered.
>
> **Threat model**: The attacker has no access to the model training process and does not know the model architecture of the target model, whose ability is weaker compared to the VQA backdoor attack. In contrast, The Dual-Key paper assumes the attacker can control the whole training process of the VQA model, which is a white-box attack scenario. In the VQA task, the poisoned samples generated by the attacker only target a certain model that the attacker wants to train. But in the image caption, the attacker's goal is to generate model-agnostic poisoned samples, which can successfully attack a variety of model architectures.
>
> **Q2**: The Figure 1 is not very intuitive. The image caption models maybe common, but involving the specific architecture in Figure 1 (The Experimental Setup Section is not clear to me) would help to understand the attack procedures. I assume the input images go into the object detector first, then the output features would be forword to ResNet101-LSTM model for text generation? Are there any other modules between object detector and ResNEt101-LSTM? Also, the proposed method adds the trigger to the center of bounding box generated by object detector. It would be better to illustrate it in the Figure 1 (for better understanding).
>
> **A2**: We apologize for the potential confusion. We have updated Figure 1 for readers to understand the process better. We give a simple explanation based on Figure 1. First, the object detector is only used to optimize the trigger pattern. After generating the trigger pattern, we add it to clean samples and modify the object names in the corresponding captions to get our poisoned samples. Then, the clean and poisoned samples are directly fed to the ResNet101-LSTM model for training. There are no other modules between the object detector and ResNet101-LSTM.
>
> **Q3**: Experiments on other CNN-LSTM and transformer based archs.
>
> **A3**: Thanks for the comments. We have added some experiments on ResNet50-LSTM, Unified VLP and ViT-GPT2. The results show that our method can still achieve high ASR on these target models. Please refer to the global comments.
>
> **Q4**: Experiments on COCO dataset.
>
> **A4**: Thanks for the comments. We have added some experiments on COCO dataset. The results show that our method can also achieve high ASR on the COCO dataset. Please refer to the global comments.
>
> **Q5**: How important is the Trigger location?
>
> **A5**: Thanks for the comments. The location of the trigger within the object is important. We set Flickr8k and ResNet101-LSTM as the dataset and the architecture, then change the trigger location to the bottom right corner and the top left corner of the object bounding box for comparisons.
>
>
> |   Location   | ASR | BLEU-4 | CIDEr | METEOR |
> | :----------: | :--: | :----: | :---: | :----: |
> | Bottom right | 51.2 | 0.208 | 0.521 | 0.224 |
> |   Top left   | 45.8 | 0.209 | 0.528 | 0.217 |
>
> From the experimental results, we observe a significant drop in ASR, this is because the corners of the bounding box don't necessarily contain objects, and it could be the background of the image. We add the trigger to the center of the bounding box, because we aim to involve the trigger in the object but without affecting the rest of the image.

---

> ### Author Response · Authors · 2023-11-21
>
> **Q6**: The definition of $ASR=\frac{N_p}{N_t}$. where $N_p$ is the number of sentences containing the target object and $N_t$
> is the number of sentences containing the ground truth source object. Here, every sample's output would be counted as one sentence, or there might be multiple sentences. In the former case, we denote $N_p$ as the number of outputs in which the target object appears and $N_t$ as the number of total samples would be the same.
>
> **A6**: Thanks for the comments. It's the former case. Every sample's output would be counted as one sentence. Your understanding is correct.

---

> > ### Comment · Reviewer_sNE8 · 2023-11-22
> > **Thanks for rebuttal**
> >
> > Thanks for the clarification and additional experiments. I have increased score to 5 for encouragement.

---

### Author Response · Authors · 2023-11-21
**Global Response**

We thank all the reviewers for their insightful reviews and constructive suggestions.

To demonstrate our method's generalizability on different architectures and datasets, we add some experiments on ResNet50-LSTM, Unified VLP [1], ViT-GPT2 and COCO dataset. And we involve CIDEr and METEOR metrics as Reviewer 1VPN suggested. For the Flickr8k and COCO datasets, we choose 'dog' and 'cat' as the source and target objects, respectively. For the Flickr30k dataset, we choose 'person' and 'toothbrush' as the source and target objects, respectively. The values in parentheses represent the difference in corresponding metrics compared with the benign model.


|  Dataset  |      Arch      | Poisoning Rate | ASR |     BLEU-4     |     CIDEr     |     METEOR     |
| :-------: | :------------: | :------------: | :--: | :------------: | :------------: | :------------: |
| Flickr8k | ResNet50-LSTM |       5%       | 84.6 | 0.210 (-0.009) | 0.538 (-0.032) | 0.221 (-0.006) |
| Flickr30k | ResNet50-LSTM |       5%       | 81.2 | 0.218 (-0.013) | 0.452 (-0.022) | 0.216 (-0.003) |
|   COCO   | ResNet101-LSTM |      2.1%      | 87.1 | 0.261 (-0.018) | 0.877 (-0.035) | 0.238 (-0.018) |
|   COCO   |  Unified VLP  |      2.1%      | 89.5 | 0.332 (-0.034) | 1.054 (-0.019) | 0.259 (-0.015) |
|   COCO   |    ViT-GPT2    |      2.1%      | 86.3 | 0.364 (-0.025) | 1.133 (-0.028) | 0.276 (-0.018) |

The results show that our method can achieve high ASR while maintaining the model's benign performance on different architectures and large dataset (COCO).

[1] Zhou et al. "Unified Vision-Language Pre-Training for Image Captioning and VQA." AAAI, 2020.